# Genetic and Epigenetic Factors of Takotsubo Syndrome: A Systematic Review

**DOI:** 10.3390/ijms22189875

**Published:** 2021-09-13

**Authors:** Valentina Ferradini, Davide Vacca, Beatrice Belmonte, Ruggiero Mango, Letizia Scola, Giuseppe Novelli, Carmela Rita Balistreri, Federica Sangiuolo

**Affiliations:** 1Department of Biomedicine and Prevention, University of Rome Tor Vergata, Via Montpellier 1, 00133 Rome, Italy; Ferradini@med.uniroma2.it (V.F.); novelli@med.uniroma2.it (G.N.); sangiuolo@med.uniroma2.it (F.S.); 2Tumor Immunology Unit, Department of Health Sciences, University of Palermo, 90134 Palermo, Italy; davide.vacca@unipa.it (D.V.); beatrice.belmonte@unipa.it (B.B.); 3Cardiology Unit, Department of Emergency and Critical Care, Policlinico Tor Vergata, 00133 Rome, Italy; ruggiero.mango@gmail.com; 4Department of Biomedicine, Neuroscience and Advanced Diagnostics (Bi.N.D.), University of Palermo, 90134 Palermo, Italy; letizia.scola@unipa.it

**Keywords:** Takotsubo cardiomyopathy (TTS), biomarkers, specific and effective treatments, TTS management, genetic and epigenetic factors, systematic review

## Abstract

Takotsubo syndrome (TTS), recognized as stress’s cardiomyopathy, or as left ventricular apical balloon syndrome in recent years, is a rare pathology, described for the first time by Japanese researchers in 1990. TTS is characterized by an interindividual heterogeneity in onset and progression, and by strong predominance in postmenopausal women. The clear causes of these TTS features are uncertain, given the limited understanding of this intriguing syndrome until now. However, the increasing frequency of TTS cases in recent years, and particularly correlated to the SARS-CoV-2 pandemic, leads us to the imperative necessity both of a complete knowledge of TTS pathophysiology for identifying biomarkers facilitating its management, and of targets for specific and effective treatments. The suspect of a genetic basis in TTS pathogenesis has been evidenced. Accordingly, familial forms of TTS have been described. However, a systematic and comprehensive characterization of the genetic or epigenetic factors significantly associated with TTS is lacking. Thus, we here conducted a systematic review of the literature before June 2021, to contribute to the identification of potential genetic and epigenetic factors associated with TTS. Interesting data were evidenced, but few in number and with diverse limitations. Consequently, we concluded that further work is needed to address the gaps discussed, and clear evidence may arrive by using multi-omics investigations.

## 1. Introduction

Takotsubo cardiomyopathy (TTS), recognized as stress’s cardiomyopathy, or as left ventricular apical balloon syndrome in recent years, is a rare pathology, described for the first time by Japanese researchers in 1990 [1]. TTS cases show acute and transient left ventricular systolic and diastolic dysfunction, which is often associated with a stressful, emotional, or physical event, and typically recovers spontaneously within days or weeks. Several stressors, including emotional or physical stress (e.g., unexpected death of a family member, suppressed terror, natural disasters or vigorous physical stress), may evoke TTS onset [2]. For example, TTS has been closely related to SARS-CoV-2 infection and the ongoing pandemic. Precisely, it has been evidenced that the huge emotional stress evocated by the SARS-CoV-2 pandemic can act as a possible TSS trigger. However, it has been reported that TTS may also represent a direct complication of SARS-CoV-2 infection [3,4], and particularly in cancer patients. Cancer patients show different risk factors (i.e., active cancer, surgery, radiotherapy, anticancer drugs), that could determine a coagulation dysfunction, characterized by an increased coagulative activity, and reciprocally by an inhibition of fibrinolysis. Consequently, individuals affected by cancer are enormously susceptible to viral infection and may have a more negative prognosis than healthy subjects. In addition, the type of cancer and related therapies, such as recent chemotherapy, radiotherapy, surgery, and other concomitant comorbidities (i.e., diabetes, cardiovascular diseases, metabolic syndrome) may increase the rate of vulnerability to the infection and its complications [4]. Furthermore, patients affected by sepsis, neurological disorders (i.e., subarachnoid hemorrhage, seizures, stroke/transient ischemic attack, cerebral tumors, head trauma), and pheochromocytoma, can also develop TTS [3,4]. Drugs, such as dopamine, dobutamine, epinephrine, norepinephrine, and anesthesia drugs, can also represent TTS triggers, evocating a TTS like disorder [5]. Likewise, it has been reported that chemotherapeutical drugs can determine TTS onset in cancer patients, as an epiphenomenon of induced cardiotoxicity [6]. Such conditions have been defined as TTS phenocopies [5], different to the classical TTS phenotypes.

In addition to negative triggers, positive life circumstances can also evocate TTS onset, as shown in recent years [2]. This has recently led to the definition of some TTS forms as happy heart syndrome [2]. It has also been observed that only some individuals develop TTS, even if exposed to the same stressors [2,5].

TTS diagnosis is now routinely based on the evidence of acute chest pain, electrocardiographic changes, troponin elevation, unobstructed coronary arteries, and a characteristic profile of circumferential left ventricular wall motion abnormalities commonly involving the apical and midventricular left ventricular walls [7]. Accordingly, TTS has been classified as a myocardial injury, but not as infarction, in the recent Fourth Universal Definition of Myocardial Infarction [7]. However, as mentioned above, only some individuals develop TTS, even if exposed to same stressor/stressors [1,5], and only some cases show severe acute complications, including 4% to 5% mortality related to cardiogenic shock and cardiac arrest [8,9]. Thus, an interindividual heterogeneity in onset and progression characterizes TTS. Despite this, TTS is typically described in postmenopausal women. Accordingly, in Western countries, a female prevalence has been observed, with a female-to-male ratio of 9:1 [2,5]. Up to 90% of all patients with TTS are, indeed, women, mostly postmenopausal. Estrogen deficiency is considered as a predisposing factor, a prerequisite [2,5,10], or better a potential mechanism involved in TTS pathophysiology, as recently demonstrated. The deprivation of estrogens has been showed to contribute to endothelial dysfunction [2,5,10]. In Japan, men are more affected than women, for reasons not completely identified [11]. However, differences in the genetic background or stressful environments of the Japanese population compared to other populations might be potential causes.

In their complexity, these observations suggest that TTS is an enigmatic disease with a multifactorial and unresolved pathogenesis. Accordingly, TTS risk factors and pathophysiological cellular and molecular mechanisms remain unclear [2,5,10]. This limited knowledge has also hampered the proper development of specific and effective TTS treatments, that are lacking until now. Thus, it is imperative to understand TTS pathogenesis and pathophysiology. This might permit understanding of typical TTS features, and identification of involved pathways and targets to use in the development of specific and effective therapies, as well as personalized treatments. However, some groups of researchers have provided recent evidence on the mechanisms and pathways involved, briefly reported in the next paragraphs. In particular, the group of Professor Camici has suggested a flowchart showing TTS pathogenesis, emphasizing a close interplay among triggers (i.e., stressors, but also other factors, as mentioned), pathogenetic factors acting as predisposing factors, mechanisms of cardiac injury, and clinical consequences (see the next paragraphs) [5].

Here, we want to shed light on potential TTS predisposing factors, by revisiting through a literature search the genetic and epigenetic factors which may eventually influence TTS pathogenesis and prognosis and justify its typical heterogeneity. Familial forms of TTS have been described [12]. However, a systematic and comprehensive characterization of the genetic or epigenetic factors significantly associated with TTS does not exist. Thus, a genetic predisposition might be involved, even if the current study is based only on few familial TTS cases [13]. In addition, promising loci, copy number variations, and polymorphisms have already been identified by genetic studies in TTS patients, but with inadequate results, requiring further investigation.

## 2. New Insights into TTS Pathophysiology: Mechanisms and Pathways from Current Evidence

Mechanisms and pathways involved in TTS pathophysiology are briefly described, using evidence from current literature.

### 2.1. Role of Catecholamines

The flowchart proposed by Camici’s group evidences that TTS can start from biological effects evoked by diverse stressors and the combined action of predisposing genetic factors. Their summary effect can evoke activation of both central and autonomic nervous systems. This [5] determines, on the one hand, the elevation of catecholamine levels able to induce myocardial injury, and on the other hand, cardiac sympathetic alterations, which contribute to modification of the left ventricular gradient of contractility. Both principal events lead to microvascular spasm and boosting of myocardial oxygen consumption, which can provoke myocardial ischemia, acute left ventricular dysfunction, and activation of survival pathways and stunning. This scenario underlines an event that contradistinguishes the physiological response to sympathetic activation in healthy individuals, physiologically determining vasodilation through activation of β2-adrenoceptors. Precisely, in TTS cases, cardiac sympathetic activity has occurred and observed to stimulate coronary microvascular constriction, because of the presence of endothelial dysfunction, even if mediated by both α1- and α2-adrenoceptors. Endothelial dysfunction also characterizes TTS, and drives different events by inducing an increased expression of both α1- and α2-adrenoceptors, as well as microvascular constriction, which can synergically cause myocardial ischemia [5].

Recently, a mechanism has been evidence that clarifies, in part, how the catecholamines evoke the pathological conditions related to TTS onset. A group of researchers from Xuzhou Medical University and the National Heart and Lung Institute of London [14] focused on the potential effects and mechanisms evoked by the recently discovered G protein-coupled estrogen receptor (GPER), in conditions of epinephrine (Epi)-induced cardiac stress, and by using a high dose of Epi and an adult rat and human-induced pluripotent stem cell-derived cardiomyocytes (hiPSC-CMs) model of study. Their results have demonstrated that GPER activation determines both a rise in left ventricular internal diameter at end-systole, and the reduction of both ejection fraction and increase in Epi caused cardiomyocyte shortening [14]. Furthermore, a moderated induced-Epi heart injury has been evidenced, by detecting the levels of plasma brain natriuretic peptide and lactate dehydrogenase release into the hiPSC-CMs culture supernatant. It has also been assessed that these molecules prevent the elevation of phosphorylation and internalization of β2-adrenergic receptors (β2AR) [14]. Thus, it has been suggested that GPER has a defensive role against TTS onset, through a balance of the coupling of β2AR and the Gαs and Gαi signaling pathways. On the other hand, it has been largely demonstrated that estrogens mediate the beneficial effects on the cardiovascular system, such as vasodilation and vascular protection, as well as on cardio-pathological conditions, including atherosclerosis and endothelial dysfunction [15].

### 2.2. Dysregulation of Metabolic Pathways

Currently, it has also been demonstrated that Takotsubo myocardium is characterized by a dysregulation of glucose and lipid metabolic pathways. In turn, this energetic deficit determines a malfunctioning Ca^2+^ metabolism, inflammation, and overexpression of remodeling pathways [16]. Cotemporally, it has been shown that an increase in reactive oxygen species (ROS) followed by the downstream cascade’s effects characteristic TTS development [17], probably as a consequence of the above-mentioned dysregulation. Increased production of ROS may determine transient coronary and peripheral endothelial dysfunction, which can cause a transient damage to myocardial contraction due to stunning (apical ballooning) [18]. Consistent with these data, it has also been demonstrated that the chronic inhibition of the expression of PI3K/AKT/mTOR pathways may have protective effects on myocardial dysfunction in TTS rats, reducing mitochondrial ROS and oxidative stress-induced apoptosis. Consequently, this suggests that the use of PI3K/AKT/mTOR inhibitors might represent a treatment for cardiovascular dysfunction induced by TTS.

### 2.3. Inflammation

As mentioned, inflammation is also involved in TTS development [19,20]. The presence of myocardial edema, necrosis, and fibrosis in TTS patients identified by using cardiac magnetic resonance (CMR) imaging confirms the presence of inflammation. Immunohistochemical staining of bioptic samples has detected an infiltration of neutrophils and macrophages [19,20]. Precisely, in human postmortem biopsies from TTS cases after onset (first 5 days), an increased CD68 + cells presence has been observed after immunohistochemical staining, suggesting that aggregates of macrophages accumulate in the myocardium [19,20]. Furthermore, investigations in TTS rats have revealed that the catecholamine-mediated TTS-like cardiac dysfunction determines the M1 (proinflammatory) polarized macrophages infiltration [19,20]. M1 macrophages are specialized in producing proinflammatory mediators, such as extra-cellular matrix degradation enzymes, cytokines, chemokines, etc. Increased expression of Toll-like receptors and apoptotic molecules have been also observed in TTS tissues [21].

In addition to tissue inflammation, systemic inflammation has also been observed in TTS cases both for short and long periods [22]. Diverse sources can evoke systemic inflammation, and among these, myocardial inflammation is considered the principal starting point, as demonstrated by emerging evidence. Furthermore, systemic inflammation has been suggested to mediate diverse mechanistic, prognostic, and therapeutic implications, as stressed recently by Yalta et al. [22].

However, additional research efforts are necessary to verify, disprove, improve, and/or combine this evidence, and provide clear information on all the mechanisms and pathways involved in TTS pathophysiology. We report in Figure 1 all the pathways described, in order to encourage other studies.

### 2.4. Role of Genetics in TTS Pathophysiology

Recently, a role of genetics in TTS pathophysiology has also been evidenced. Camici et al. have underlined a closed interplay between stressors and pathogenetic factors [5]. In addition, the Limongelli et al. have encouraged genetic studies and stressed a genetic predisposition in TTS [12,13].

However, there are no clear data in the literature, and our interest in this review is in explicating this aspect by performing a systematic literature search, as extensively reported below. In addition, another aspect that we want to evidence is the eventual close relationship between the above-described pathways, discovered to be involved in TTS pathophysiology, and the potential associations of genetic variants in genes codifying these molecules, with susceptibility, protection against, and severe prognosis of TTS.

## 3. Methods

### Search Strategy and Study Selection

To evidence the genetic factors related to TTS, a comprehensive search from the international web databases PubMed, Medline, and Web of Science, including articles published before June 2021, has been conducted. The following terms have been used as search keywords without restrictions on language, ethnicity, or geographic area: “Takotsubo gene”, “Takotsubo genes”, “stress-induced Takotsubo Syndrome”, “Takotsubo Syndrome genes”, “Takotsubo Syndrome genetic”, “Polymorphisms Takotsubo” and “stress cardiomyopathy”. This strategy has led to the data described in the subsequent paragraphs (Figure 2).

## 4. Results and Discussion

### 4.1. Genetic Studies on Familial Forms of TTS

A restricted number of cases affected by familial TTS form has been reported in the literature. A genetic predisposition for the development of TTS has also been suspected based on existing case reports of this disease among siblings and mother–daughters. Multiple cases of TTS in the same family and recurrent cases indicate a genetic component.

Data are described in Table 1 [23,24,25,26,27,28,29,30].

### 4.2. GWAS and Polymorphism Studies

Current research has investigated possible associations between TTS and single nucleotide polymorphisms (SNPs) in genes involved with sympathetic stress. Various studies have been published describing SNPs potentially implicated in the pathogenesis of TTS (Table 2) [31,32,33,34,35,36,37,38,39,40,41,42,43].

Among these, the genes encoding the B1 (*ADRB1*), B2 (*ADRB2*) and alpha 2c (*ADRA2C*) adrenergic receptors have been examined, due to the involvement of their variants in functionally modulating the cardiac response to catecholamines. Precisely, in a cohort of 41 TTS patients, Sharkey et al. [31] did not find any difference in the distribution frequency of adrenoceptor *ADRA2C* and adrenoceptor *ADRB1* polymorphisms if compared with controls. The same polymorphisms were evaluated in another cohort of 61 patients [32], reporting a different distribution of *ADRB1* (rs1801253) p.Arg389Gly SNP. To be exact, homozygous Arg/Arg is more frequent in TTS, and *ADRB2* (rs1042714 p.Gln27Glu) homozygous Gln/Gln is more frequent in healthy controls. However, no significant differences for B2-adrenergic receptor (rs1042713, p.Arg16Gly) variation were detected.

Spinelli et al. [33] investigated the eventual associations of genetic polymorphisms in *ADRB1*, *ADRB2*, *GNAS*, *GRK5* genes with TTS, enrolling 22 cases. The genetic analysis showed a similar distribution between cases and controls for the major number of examined polymorphisms. The unique significant difference was assessed for the rs17098707 polymorphism in the *GRK5* gene, with a higher prevalence of the leucine allele at position 41 in TTS subjects. In contrast with this paper, Figtree et al. [34] detected no association. Successively, the Australian Tako-tsubo Taskforce enrolled 92 TTS patients [34] to be genotyped for rs17098707 in *GRK5* gene, rs1801253 in *ADRB1* gene, rs1800888 in *ADRB2* gene, rs4680 in *COMT* gene, rs6915267 and rs71017521 in *ESR1* gene. No associations were found.

In 2015, the rs17098707 in *GRK5* was also studied by the Novo group in 20 TTS patients and 22 healthy individuals [35]. While the AA genotype was detected in 12 TTS patients, the ‘variant’ T allele was detected only in five patients in the hetero-zygositic state (AT) and only in three patients in the homo-zygositic state (TT). In the wild type control group, the frequency of the AA genotype was observed in the majority of the analysed population (92%).

During the same year, Vriz et al. [36] analysed β1- and/or β2-adrenoceptor polymorphisms (β1-Arg389Gly polymorphism; rs1801253 *ADRB1*, β2-Arg16Gly polymorphism; rs1042713 ADRB2, β2-Gln27Glu polymorphism; rs1042714 ADRB2) in 97 TTS patients compared with 81 subjects showing anterior STEMI (acute ST-elevation myocardial infarction) and with a cohort of 101 controls. The frequency of all three genotypes is distributed in a way significantly different between the groups, when comparing TTS patients and controls. On the contrary, no differences were evidenced when TTS and STEMI patients were compared.

Citro et al. evaluated [37] a population of 29 TTS patients and more than 1000 healthy controls for the presence of variants in the protein Bcl2- associated athanogene 3 (BAG3). Concerning SNP rs35434411 (R71Q), two TTS patients were found heterozygous, while all analyzed controls (*n* = 1043) were homozygous for the major allele (*p* = 0.0007). In addition, two TTS patients (7%) were homozygous for SNP rs3858340 (P407L) of *BAG3*, versus 1% of analysed controls (*p* = 0.045). Yet no statistical differences in the frequency of SNP rs2234962 (C151R) between patients and controls was also detected. This study was extended by sequencing exons 2–4 of the coding sequence and the entire 3′-UTR of *BAG3* in a total of 70 women TTS patients [38]. Among the genomic variants identified, a particularly frequent variant in the 3′UTR (rs8946) was identified: 62.8% of TTS patients carried this nucleotide variant (12.8% were homozygous), while the variant was present in only 45.6% of the samples of the control group (7.4% were homozygous). Since miR-371-5p binds to *BAG3* 3′UTR region, a novel posttranscriptional epi-induced mechanism may be hypothesized for which *BAG3* 3′UTR variants can impair miR-371-5p activity.

In 2018, by genotyping a large TTS cohort, Mattson et al. [39], demonstrated a lack of association of candidate SNPs with TTS in the *ADRB1*, *GRK5* and *BAG3* genes, previously suggested to contribute to this disease. Precisely, 1438 samples (461 TTS patients, 403 controls with CAD and 574 healthy controls) were genotyped for the following polymorphisms: rs1801253, rs2230345 and rs8946 in *ADRB1*, *GRK5* and *BAG3* genes, respectively. The genotype distribution did not differ among the three study groups.

In 2016, Cecchi et al. [40], analyzed 75 TTS women compared to acute coronary syndrome (ACS) patients and control subjects for the detection of the G1691A polymorphism in the factor V gene (factor V Leiden) and the G20210A polymorphism in the factor II gene. For both variants no significant difference in their distribution was detected.

Successively, a genotyping study was conducted evaluating the SNPs in genes encoding estrogen receptors, since literature data demonstrate a correlation between myocardial infarction and genetic variants in genes of *ESR1* and *ESR2* oestrogen receptors. Two variants (rs2234693 and rs9340799) in the *ESR1* gene and two (rs1271572 and rs1256049) in the *ESR2* gene were examined in 81 consecutive white women: 22 with TTS, 22 with acute myocardial infarction and 37 asymptomatic healthy controls [41]. Women carrying the T allele at the rs2234693 locus of the *ESR1* gene and carrying a T allele at the rs1271572 locus of the *ESR2* gene had an even higher risk of occurrence of TTS.

Since the studies on polymorphisms did not reveal significant data, some groups focused on genome-wide association study (GWAS) or Whole exome sequencing (WES). GWAS is an approach used in genetics research to associate specific genetic variations with specific diseases. The method involves scanning the genomes from many different people and looking for genetic markers that can be used to predict the presence of a disease. Precisely, GWAS analysis was conducted in 96 TTS patients (91 females, five males) and 475 healthy controls [42] showing 68 promising candidate loci. In 18 (rs12612435, rs7070797, rs9392780, rs13273616, rs1154275, rs72970558, rs62253104, rs4676168, rs4961212, rs6944978, rs4812257, rs162487, rs4605019, rs113154180, rs13179382, rs12444925, rs17146144 and rs56403110), the top SNPs supported by SNPs in high Linkage Disequilibrium (r > 0.8) and *p* < 10^−3^.

Exome sequencing may be used for identifying causal variants of rare disorders by using the Next Generation Sequencing (NGS) technology. Exome analysis, genotyping array analysis, and array comparative genomic hybridization were carried out on 28 Christchurch EqSCM cases [43]. Moreover, WES and Cardio-MetaboChip genotyping analyses were performed without determining evident roles for exonic mutations in a monogenic model, or for polymorphisms in a polygenic model.

### 4.3. Sequencing of Takotsubo Cohort

In 2007, in the female TTS patients, a cDNA microarray study was performed comparing the expression of left ventricular biopsies taken in the acute phase (group A) and an after functional recovery (group B) [44]. Results were confirmed validating selected genes by means of real-time PCR and immunohistochemistry. This study demonstrated a significant contribution of oxidative stress to the pathophysiology of TTS, probably released by an excess of catecholamine. Consequently, a potential compensatory mechanism could be induced as a boost of protein biosynthesis and an activation of cell survival cascade [44].

In 2009, the same group demonstrated that TTS is associated with alteration of Ca2þ-handling proteins, playing a pivotal role in contractile dysfunction. In fact, they described alterations of the expression levels of Ca2þ-regulatory proteins assessed by real-time PCR, western blot, and immunohistochemistry in 10 TTS patients [45].

In 2009, Kleinfeldt et al. reported for the first-time data on a female individual with TTS, who happened to be carrier of an *FMR1* gene mutation, with alleles of an intermediate size between 40–55 triplet pre-mutations [46].

In 2014, Goodloe et al. [47] used exome sequencing for comprehensive genotyping of a TTS cohort (28 TTS subjects, including a mother–daughter pair), enabling investigation of 486 genes of a pathway network related to adrenergic signaling. Two-thirds of TTS cases carried more than one filtered adrenergic pathway variant, and 11 genes harboured a variant in ≥2 cases. The mother–daughter pair shared missense variants located in highly conserved functional domains of *ADH5*, *CACNG1*, *EPHA4*, and *PRKCA* genes. No common genes were detected when analyzing independently from the adrenergic pathway. Thus, these results further highlight genetic heterogeneity in TTS susceptibility, also suggesting a polygenic inheritance in which adrenergic pathway dysregulation plays a cumulative effect in a subset of individuals.

In 2016, an exome sequencing study was performed on seven female sporadic TTS patients [48], evidencing in all subjects predicted deleterious variants in described cardiomyopathy genes. The variants were not described in public human genome data or in a public database associated with cardiac dysfunction.

Two years later a NGS analysis was conducted in a TTS patient showing clinical signs of an unrecognized genetic cardiomyopathy leading to the discovery of a heterozygous mutation in exon 9 of the *TTN* gene: c.1489G > T (p. E497X) [49].

Lacey et al. performed WES genotyping array analysis and array comparative genomic hybridization on 24 of the 28 Christchurch earthquake-associated TTS cases [43]. From WES, the most striking finding was the observation of a markedly elevated rate of rare, heterogeneous copy number variants (CNV) of uncertain clinical significance (in 12/28 subjects). Several of these CNVs impacted on genes of cardiac relevance including *RBFOX1*, *GPC5*, *KCNRG*, *CHODL*, and *GPBP1L1*.

In 2020, the Pan group analyzed microarray datasets GSE95368 derived from the Gene Expression Omnibus (GEO) database [50], and for the first time they identified differentially expressed genes (DEGs) between TTS and controls. Then the DEGs were used for Gene Ontology (GO) and Kyoto Encyclopedia of Genes and Genomes (KEGG) pathway enrichment analysis. Lastly, the protein–protein interaction (PPI) network was constructed and Cytoscape was used to find the key genes. Among 25 DEGs, ten were upregulated and 15 downregulated. Moreover, seven genes including *APOE*, *MFGE8*, *ALB*, *APOB*, *SAA1*, *A2M*, and *C3* were classified as hub genes of TTS, useful for both diagnostic biomarkers and for molecular targets for the treatment of TTS.

In 2021, Khurana et al. [51] tested the hypothesis that the cardioprotective benefit of Suberanilo-hydroxamic acid (SAHA) in a pre-clinical TTS study could be due to an epigenetic (acetylation/deacetylation: Ac/Dc) axis. RNA-sequencing showed that the SAHA treatment was able to modify the transcriptional activity of some genes with a shift towards cardiac benefit. These data indicate that the core pathways identified might be regulated by SAHA administration. Surprisingly, the Ac/Dc axis involved the lysine deacetylation of the SCAAR-associated genes *ADRB1*, *BAG3* and *GRK5*. The expression of stress-induced genes was influenced in the REV model (ISO/SAHA vs. Control) as well as the Ac/Dc index in the REV (ISO/SAHA) vs. ISO groups.

### 4.4. Epigenetic Factors and TTS

Recently, the role of epigenetic factors, including MicroRNAs (miRNAs/miRs) and the acetylation/deacetylation (Ac/Dc) axis (the latter as mentioned and discussed above [51]), is emerging in different pathologies. MicroRNAs (miRNAs/miRs) are a class of highly conserved, small (19–25 nucleotides) noncoding post-transcriptional regulators of proliferation, angiogenesis, differentiation, and apoptosis, but also were recently suggested as biomarkers of several cardiovascular diseases, such as heart failure, stable coronary artery disease, and acute MI [52,53,54,55,56,57]. Such evidence also led us to search, from the international web databases PubMed, Medline and Web of Science, articles published before June 2021 on circulating miRNAs associated with TTS. The following data have been reported. In 2014, the Templin group demonstrated that circulating miR-16 and miR-26 resulted in increased TTS versus STEMI or healthy controls. The *Braun* group, in 2013, evidenced that miR-1 [58] and miR-133a were largely increased in STEMI vs. TTS, even if elevated in both vs. controls. miR-1 and miR-133a are, indeed, typical of cardiac tissue, consequently they are more likely to reproduce the grade of myocardial injury than other pathological conditions. Accordingly, they were more discreetly elevated in TTS with respect to STEMI associated with levels of cardiac troponin and creatine kinase-MB in circulation [59].

No other data have been reported on miRs in TTS in the literature. However, some researchers have evaluated whether miR-16 and miR-26a, specifically increased in TTS as above described, have a close relation with TTS or simply are the result of the activation of catecholamines or their damage. In addition, the organ or tissue of origin of miR-16 and miR-26a in TTS has also been investigated, assessed only in blood samples from TTS patients. However, it is well known that miR-16 and miR-26a have been detected in circulation in stress, depression, and anxiety conditions, as documented by the results of three recent studies [60,61,62]. In addition, the Templin group has elegantly reported that TTS has a significant relationship with pre-existing psychiatric disorders, i.e., anxiety and depression [63]. Consequently, miR-16 and miR-26a could play the role of effectors in the brain–heart interaction in TTS cases. Some responses to these questions have recently arrived from a study conducted in 2021. Precisely, the researchers demonstrated that miR-16 and miR-26a reduce the TTS-like variations in the contractile system from TTS rats (in vivo and in vitro) and human cardiomyocytes, and permitted detection of the involved fundamental mechanisms [64].

Further studies on TTS-associated miRs are imperative because they can contribute to clarify their role, as well as their potential capacity to be active players for TTS onset and complications. Consequently, the evaluation of blood levels in TTS patients, clinically followed, or recovered during successive stress periods, might predict the probability of reappearance of acute episodes, which represent a significant risk in these cases, by consenting to the application of preventive strategies.

On the other hand, the knowledge of the mechanisms involved may also allow the development of other preventive pharmacological therapies, including the pre/anti-miRNA constructs which are now influencing clinical practice.

## 5. Conclusions

Clear causes of TTS, with predisposing genetic factors, have long remained uncertain, prevalently because of the restricted literature on this intriguing syndrome. However, the growing frequency of TTS cases in recent years, and particularly correlated to theSARS-CoV-2 pandemic [65,66], has led to an imperative necessity for a complete knowledge of TTS pathophysiology, and to the urgent development of specific and effective treatments. Models have been recently proposed, including the pathophysiological flowchart from the Camici group [5]. Here, we have reported and summarized, in a model, all the mechanisms and pathways described by the current literature (see Figure 1), with the aim to evidence the role of genetics in TTS. Genetics seems to have a crucial role, as particularly stressed by the Limogelli group [12,13]. Based on this evidence, we have performed this systematic review (Figure 2) to cast light on the potential genetic and epigenetic factors significantly associated with TTS. Consequently, a systematic search of published studies before June 2021 from international web databases PubMed, Medline, and Web of Science has been executed. The data obtained have led us to suggest that genetic variants of *ADRB1*, *GRK5* and *BAG3* genes, or variants of *APOE*, *MFGE8*, *ALB*, *APOB*, *SAA1*, *A2M*, and *C3* genes, significantly associated with TTS, can probably interact with the environment and emotional and physical stress, and predispose certain individuals more than others to develop TTS, earlier and with more severe complications. This might justify the heterogeneity, both of having or not having acute episodes, in showing different clinical pictures accompanied by more severe, or not, complications in TTS cases.

No significant genetic data about the pathways described in the model illustrated in Figure 1 exist until now. However, we hope this model and the data, here reported may encourage additional genetic studies, as well as other more sophisticated molecular studies by using technologies of the latest generation. Cotemporally, the proposed model might lead to further and larger mechanistic studies both in animals and human cell cultures, for identifying ulterior pathways and for clarifying the complex TTS pathophysiology.

Furthermore, we have considered the hypothesis that the data evidenced in this review might be influenced by the major limitation which generally characterizes a systematic literature revision, i.e., having its origin in retrospective studies. This enhances the probability of selection and information biases. In addition, the increase of cases during the SARS-CoV-2 pandemic, of about four–fivefold, as well as the rise of mortality rate in cases with TTS, cancer and COVID-19, or with old age (but not in nonagenarians or centenarians, as we have described in a recent review [67]), would also be considered in this reflection. In addition, the pathological conditions defined as phenocopies of TTS-like syndrome would also be evaluated.

In conclusion, we can affirm that the current available literature has not comprehensively addressed the possibility of identifying clear genetic traits facilitating TTS management, as well as the increased risk of recurrence of acute episodes in TTS cases. Consequently, future research with omics investigations [68] and involving larger populations is warranted.

The size limitation of these studies has prevented the detection of a common polymorphism effect, encouraging further studies involving an accurate selection of phenotypes, and data sharing for increasing both the number and the quality of the information to better indicate the potential role of the predisposing genetic and epigenetic TTS factors.

Finally, we also suggest that a better strategy might be the integration of multi-omics approaches as a promising tool for the identification of appropriate biomarkers, such as genetic and epigenetic risk factors, and therapeutic targets for TTS, as we have previously specified for other cardiovascular diseases [67,68]. The analysis of genetic, epigenetic, metabolomic, microbiomic, and nutrigenomic profiles could be encouraging, and lead to the identification of promising biomarkers and personalized medicine in the near future. Such analyses could also lead to an understanding of the impact of gender differences in TTS pathophysiology in the diverse populations studied.

## Figures and Tables

**Figure 1 ijms-22-09875-f001:**
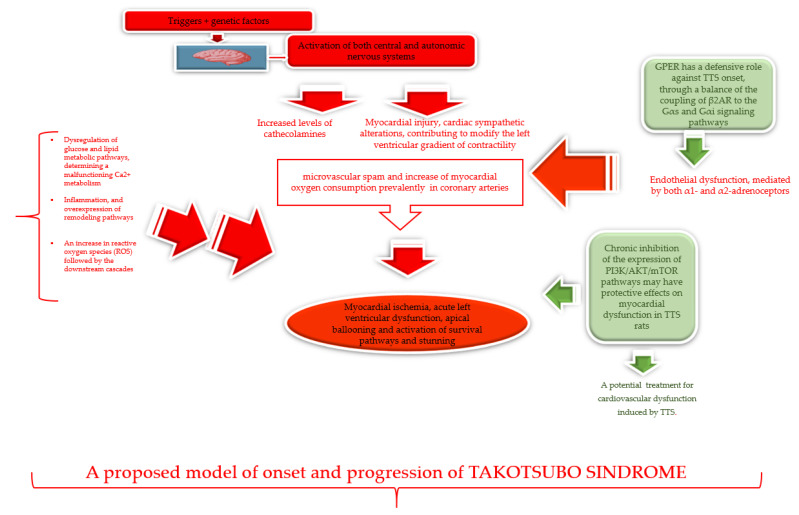
A proposed model of TTS pathophysiology from current evidence: in red the pathways involved, and in green the pathways able to delay or escape the onset and progression of pathological conditions leading to TTS and its related complications.

**Figure 2 ijms-22-09875-f002:**
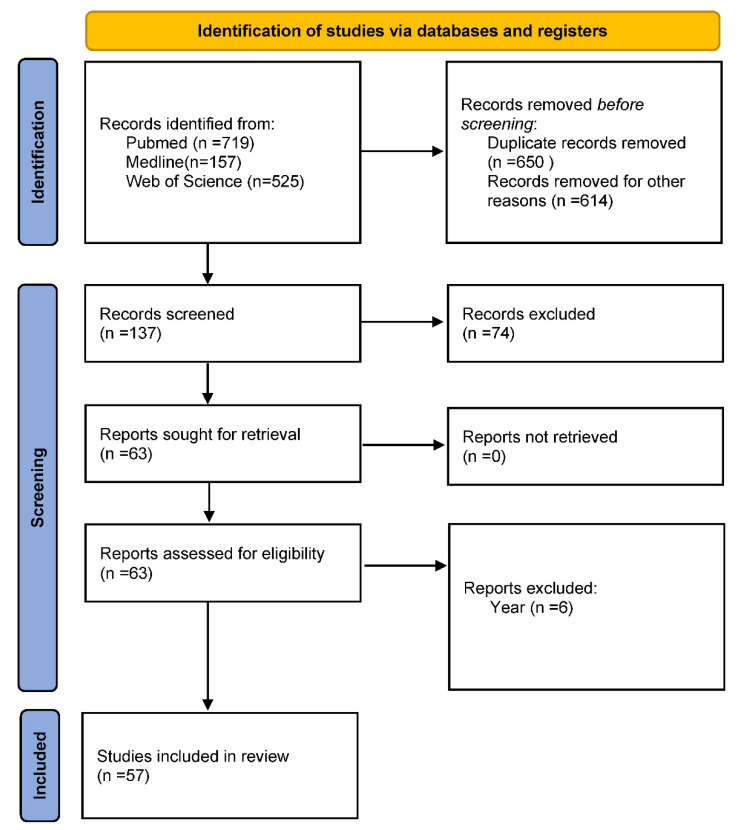
PRISMA flow diagram. Description of the search strategy and exclusion/inclusion criteria.

**Table 1 ijms-22-09875-t001:** Familial case of Takotsubo.

Authors	Year	Patients
**Pison**	2004	two sisters
**Kumar**	2010	mother and daughter
**Schultz**	2012	In three families, several close relatives developed SIC
**Subban**	2012	A 68-year-old female and her daughter aged 43 got admitted to our institute simultaneously
**Musumeci**	2013	At the same time in two sisters
**Ikutomi**	2014	two sisters
**Caretta**	2015	two sisters
**Ekenback**	2019	twin sisters

**Table 2 ijms-22-09875-t002:** Summary of polymorphisms found in patients affected by Takotsubo cardiomyopathy.

Reference	Year	Population	Analyzed Variant	Evidence
**Sharkey**	2009	41 female TTS and 43 female controls	*ADRB1* (amino acid positions 389 and 49) and *ADRA2C* (deletion 322_325)	Genotype polymorphism frequencies are not significantly different in TTS patients compared to controls
**Spinelli**	2010	22 TTS and 740 control	rs35230616 and rs1801253 for *ADRB1*; rs1042713, rs1042714, and rs1800888 for *ADRB2*; rs11554276 for *GNAS*; rs17098707 and rs34679178 for *GRK5*	The prevalence of polymorphisms of *ADRB1*, *ADRB2*, and *GNAS* were similar between patients and controls. Conversely, the percentage of patients who presented rs17098707 polymorphism of the *GRK5* gene was significantly higher.
**Vriz**	2011	61 TTS and 109 controls	rs1801253 for *ADRB1*, rs1042713 and rs1042714 for *ADRB2*	The rs1801253 for *ADRB1* genotype frequencies were significantly different in the two groups.
**Citro**	2013	29 TTS and 1043 controls	*BAG3* polymorphisms: R71Q (rs35434411); C151R (rs2234962); P407L (rs3858340)	Two TTC patients were heterozygous for SNP rs35434411, whereas all analyzed controls were homozygous for the major allele
**Figtree**	2013	92 TTS	rs17098707 for GRK5, rs1801253 for *ADRB1*; rs1800888 for *ADRB2*; rs4680 COMT; rs6915267 and rs71017521 for *ESR1*	They have found no association of genetic variants in the *ESR1*, *ADRB1*, *ADRB2*, or *COMT* and *GRK5* genes
**D’Avenia**	2015	70 TTS women and 81 healthy donors	They sequenced exons 2–4 of the coding sequence and the entire 3′UTR of *BAG3*.	Mutations and polymorphisms detected in the *BAG3* gene included a frequent nucleotide in the 3′-UTR region of TTS patients, resulting in loss of binding of microRNA-371a-5p
**Novo**	2015	20 TTS	Analysis of the L41Q polymorphism of *GRK5*	They found a significant difference in the frequency of *GRK5* polymorphism between TTS patients and controls
**Vriz**	2015	97 TTS, 81 with anterior STEMI and 101 controls	rs1801253 for *ADRB1,* rs1042713 and rs1042714 for *ADRB2*	In a TTS cohort compared with anterior STEMI patients, β-adrenoceptor polymorphisms were similar. β-Adrenoceptor polymorphisms in TTS patients differed from normal subjects
**Cecchi**	2016	75 age- and sex-matched acute coronary syndrome (ACS) patients, both enrolled during the acute phase, and in 75 control subjects	factor II (G20210A) and V (G1691A) polymorphisms	No significant difference between the three groups were observed
**Eitel**	2017	96 TTS and 475 healthy controls	A genome-wide association study (GWAS)	The results of GWAS analysis showed several promising candidate 68 loci
**Lacey**	2017	28 TTS women	exome analysis, genotyping array analysis, and array comparative genomic hybridization	No mutations or SNPs detected for monogenic or polygenic model respectively
**Pizzino**	2017	81 TTS women	rs2234693 and rs9340799 in *ESR1* gene; rs1271572 and rs1256049 in *ESR2*	The study reports preliminary findings suggesting a possible link between ESR polymorphisms and the occurrence of TTS.
**Mattson**	2018	461 TTS, 403 controls with CAD and 574 controls without CAD.	rs1801253 for *ADRB1*, rs2230345 for *GRK5* and rs8946 for *BAG3*	By genotyping a TTS cohort, they demonstrate a lack of association between candidate SNPs in the *ADRB1, GRK5* and *BAG3* genes, earlier suggested to contribute to TTS.

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
