# Peer review of "Genetic and Epigenetic Factors of Takotsubo Syndrome: A Systematic Review"

_ijms, 2021, doi:10.3390/ijms22189875_

Round 1

Reviewer 1 Report

The topic of the review if potentially very interesting, however the authors needs to  improve the manuscript for publication. Instead that provide just a list of the papers published on genetics and genomics of Takotsubo cardiomyopathy, I would suggest a more critical approach to the literature and include a figure on possible mechanisms for the disease.

The paragraph on page 5-6, lines 187-195 and the paragraph on page 7, lines 233-237ù, are practically the same although one is presented in the "GWAS and polymorphism studies" and the other in "Sequencing of Takotsubo cohort". The authors should be more clear and separate descriptive studies from more mechanistically oriented studies.

I would also suggest the authors to look if more mechanistic studies have been conducted in cell or animal models which would be more interesting for the readers of IJMS.

Reviewer 2 Report

Ferradini ed al, through a detailed research on scientific literature, wrote this review on Takotsubo syndrome (TTS) focusing their attention on genetic and epigenetic factors influencing TTS pathogenesis. In order to do that, the authors considered the articles available on international web databases such as PubMed, Medline and Web of Science that has been published before June 2021.

Although the extensive revision of the literature, the issue of genetic and epigenetic factors in TTS remains vague for both clinical and research corners. So that, although well written, the present review, in my opinion, is not interesting for the readers of International Journal of Molecular Sciences.

Reviewer 3 Report

The manuscript titled " Genetic and Epigenetic factors of Takotsubo syndrome: a systematic review" is a very interesting review describing Takotsubo syndrome in several diseases with a particular focus on COVID-19. The manuscript is well written and the overall structure is well performed. References are updated and of good quality. However, authors should:

1) Describe the possible role of Takotsubo as epiphenomenon of cardiotoxicity in cancer patients ( you can cite doi: 10.1097/FJC.0000000000001026. )

2) Perform a more appropriate description of the cardiovascular events and coagulation dysfunction in patients with COVID-19 ( you can cite  doi: 10.3390/cancers12113316. )

3) Describe the pathophysiology of Takotsubo syndrome and possible biochemical pathways involved ( in the introduction)

The mansucript will be acceptable after minor revision

Round 2

Reviewer 1 Report

Major comments:

The manuscript needs major improvement in the English presentation

I would shorten the introduction and divide the paragraph “New insights in the TTS pathophysiology: mechanisms and pathways from current evidence” in subparagraphs: role of catecholamines, Dysregulation of metabolic pathways and Inflammation.  

The figure 1 is very similar to the one of the paper referenced in 5. The authors should at least add the sentence: “modified from”

Line 94: I would suggest a possible mechanism for the increased prevalence of TTS in males: for example, the genetic background, or differences in stressful environments as compared with other populations…..

Minor comments:

Line 26: Accordingly, familial form of TTS has been described” should be: “Accordingly, familial forms of TTS have been described”.

Line 50: “it reports, that TTS also represents” should be : It has been reported that TTS may also represent”

Line 60: “may increase the rate of vulnerability” I would add “may increase the rate of vulnerability to the infection and its complications”  

Line 63: the sentence needs the references.

Line 65: the sentence needs the references.

Line 65: “Likewise, it reports” should be: “Likewise, it has been reported”. Furthermore, the sentence needs references on line 67

Line 71: the sentence needs the references.

Line 71-72: “This has recently led to define TTS as happy heart syndrome.” Would be better as: “This has recently led to define some forms of TTS as happy heart syndromes.”

Line 74: the sentence needs the references.

Line 76: “performed thanks to” should be: “based on the evidence of…”

Lines 79-80: “apical and midventricular myocardium.” Better if “apical and midventricular Left ventricular walls”

Lines 85-87: “Thus, an interindividual heterogeneity in onset and progression characterizes TTS, as typical is the strong predominance of TTS in postmenopausal women” I would change to: “Thus, an interindividual heterogeneity in onset and progression characterizes TTS. Despite this heterogeneity, TTS is typically described in postmenopausal women”.

Lines 85-90: the sentences need the references.

Lines 93-94: “In Japan, men are, while, more affected than women, for unknown reasons” I would delete “while”

Lines-95-98: “Concerning the TTS treatments, they have been not established by European Society of Cardiology (ESC), because of absent of a precise understanding on molecular and cellular mechanisms involved in TTS pathophysiology until now” Please rephrase checking for grammar. Furthermore why do the authors just cite the ESC and not AHA and ACC?

Line 209: I would delete “Another aspect to clarify:”

Author Response

CommentsReviewers.

We thank the Reviewers for their appropriate and constructive comments. We took into great considerations the criticisms, and particularly those from reviewer 1# and we feel we addressed all the raised concerns. We believe that our manuscript is significantly improved by following the Reviewers’ suggestions.

Our itemized responses to the each of the Reviewers’ comments follow.

Comments from the editors and reviewers:
-Reviewer 1

Comments and Suggestions for Authors

Major comments:

The manuscript needs major improvement in the English presentation

I would shorten the introduction and divide the paragraph “New insights in the TTS pathophysiology: mechanisms and pathways from current evidence” in subparagraphs: role of catecholamines, Dysregulation of metabolic pathways and Inflammation.  

1.Author comment. We thank the Reviewer 1 for his/her important suggestions, which have helped us in improving all the sections of our review. Concerning the observation on English presentation of our manuscript, significant improvements have been performed in the text of the Introduction section, subsequent paragraphs, and the conclusion section. In addition, we revised introduction, by reducing it and adding another appropriate section with short paragraphs, for describing the pathways and mechanisms involved in TTS pathophysiology, as suggested (see lines 37-124; 129-242, pgg1-6).

The figure 1 is very similar to the one of the paper referenced in 5. The authors should at least add the sentence: “modified from”

2.Author comment. As suggested, we have modified the Figure 1.

Line 94: I would suggest a possible mechanism for the increased prevalence of TTS in males: for example, the genetic background, or differences in stressful environments as compared with other populations…..

3.Author comment. We thank the Revisor 1 for the suggested mechanism, which has been reported in the sentence. Precisely, we have reported: In Japan, men are more affected than women, for reasons not completely identified [11]. However, differences in the genetic background or stressful environments of Japanese population than other populations might be the potential causes (see lines 94-97, pg 3).   

Minor comments:

Line 26: Accordingly, familial form of TTS has been described” should be: “Accordingly, familial forms of TTS have been described”.

Line 50: “it reports, that TTS also represents” should be : It has been reported that TTS may also represent”

 Line 60: “may increase the rate of vulnerability” I would add “may increase the rate of vulnerability to the infection and its complications”  

Line 63: the sentence needs the references.

Line 65: the sentence needs the references.

Line 65: “Likewise, it reports” should be: “Likewise, it has been reported”. Furthermore, the sentence needs references on line 67

Line 71: the sentence needs the references.

Line 71-72: “This has recently led to define TTS as happy heart syndrome.” Would be better as: “This has recently led to define some forms of TTS as happy heart syndromes.”

Line 74: the sentence needs the references.

Line 76: “performed thanks to” should be: “based on the evidence of…”

Lines 79-80: “apical and midventricular myocardium.” Better if “apical and midventricular Left ventricular walls”

Lines 85-87: “Thus, an interindividual heterogeneity in onset and progression characterizes TTS, as typical is the strong predominance of TTS in postmenopausal women” I would change to: “Thus, an interindividual heterogeneity in onset and progression characterizes TTS. Despite this heterogeneity, TTS is typically described in postmenopausal women”.

Lines 85-90: the sentences need the references.

Lines 93-94: “In Japan, men are, while, more affected than women, for unknown reasons” I would delete “while”

4.Author comment. We thank the Revisor 1 for all the changes above suggested. We have performed all the corrections as indicated in text in blue ( see lines 37-124; 129-242, pgg1-6).

Lines-95-98: “Concerning the TTS treatments, they have been not established by European Society of Cardiology (ESC), because of absent of a precise understanding on molecular and cellular mechanisms involved in TTS pathophysiology until now” Please

rephrase checking for grammar. Furthermore why do the authors just cite the ESC and not AHA and ACC?

5.Author comment. We have rephrased and improved grammar quality of the indicated sentences  (see lines-98-106, pg. 3).

Line 209: I would delete “Another aspect to clarify:”

6..Author comment. We have changed the title of this paragraph.

CommentsReviewers.

We thank the Reviewers for their appropriate and constructive comments. We took into great considerations the criticisms, and particularly those from reviewer 1# and we feel we addressed all the raised concerns. We believe that our manuscript is significantly improved by following the Reviewers’ suggestions.

Our itemized responses to the each of the Reviewers’ comments follow.

Comments from the editors and reviewers:
-Reviewer 1

Comments and Suggestions for Authors

Major comments:

The manuscript needs major improvement in the English presentation

I would shorten the introduction and divide the paragraph “New insights in the TTS pathophysiology: mechanisms and pathways from current evidence” in subparagraphs: role of catecholamines, Dysregulation of metabolic pathways and Inflammation.  

1.Author comment. We thank the Reviewer 1 for his/her important suggestions, which have helped us in improving all the sections of our review. Concerning the observation on English presentation of our manuscript, significant improvements have been performed in the text of the Introduction section, subsequent paragraphs, and the conclusion section. In addition, we revised introduction, by reducing it and adding another appropriate section with short paragraphs, for describing the pathways and mechanisms involved in TTS pathophysiology, as suggested (see lines 37-124; 129-242, pgg1-6).

The figure 1 is very similar to the one of the paper referenced in 5. The authors should at least add the sentence: “modified from”

2.Author comment. As suggested, we have modified the Figure 1.

Line 94: I would suggest a possible mechanism for the increased prevalence of TTS in males: for example, the genetic background, or differences in stressful environments as compared with other populations…..

3.Author comment. We thank the Revisor 1 for the suggested mechanism, which has been reported in the sentence. Precisely, we have reported: In Japan, men are more affected than women, for reasons not completely identified [11]. However, differences in the genetic background or stressful environments of Japanese population than other populations might be the potential causes (see lines 94-97, pg 3).   

Minor comments:

Line 26: Accordingly, familial form of TTS has been described” should be: “Accordingly, familial forms of TTS have been described”.

Line 50: “it reports, that TTS also represents” should be : It has been reported that TTS may also represent”

 Line 60: “may increase the rate of vulnerability” I would add “may increase the rate of vulnerability to the infection and its complications”  

Line 63: the sentence needs the references.

Line 65: the sentence needs the references.

Line 65: “Likewise, it reports” should be: “Likewise, it has been reported”. Furthermore, the sentence needs references on line 67

Line 71: the sentence needs the references.

Line 71-72: “This has recently led to define TTS as happy heart syndrome.” Would be better as: “This has recently led to define some forms of TTS as happy heart syndromes.”

Line 74: the sentence needs the references.

Line 76: “performed thanks to” should be: “based on the evidence of…”

Lines 79-80: “apical and midventricular myocardium.” Better if “apical and midventricular Left ventricular walls”

Lines 85-87: “Thus, an interindividual heterogeneity in onset and progression characterizes TTS, as typical is the strong predominance of TTS in postmenopausal women” I would change to: “Thus, an interindividual heterogeneity in onset and progression characterizes TTS. Despite this heterogeneity, TTS is typically described in postmenopausal women”.

Lines 85-90: the sentences need the references.

Lines 93-94: “In Japan, men are, while, more affected than women, for unknown reasons” I would delete “while”

4.Author comment. We thank the Revisor 1 for all the changes above suggested. We have performed all the corrections as indicated in text in blue ( see lines 37-124; 129-242, pgg1-6).

Lines-95-98: “Concerning the TTS treatments, they have been not established by European Society of Cardiology (ESC), because of absent of a precise understanding on molecular and cellular mechanisms involved in TTS pathophysiology until now” Please

rephrase checking for grammar. Furthermore why do the authors just cite the ESC and not AHA and ACC?

5.Author comment. We have rephrased and improved grammar quality of the indicated sentences  (see lines-98-106, pg. 3).

Line 209: I would delete “Another aspect to clarify:”

6..Author comment. We have changed the title of this paragraph.

CommentsReviewers.

We thank the Reviewers for their appropriate and constructive comments. We took into great considerations the criticisms, and particularly those from reviewer 1# and we feel we addressed all the raised concerns. We believe that our manuscript is significantly improved by following the Reviewers’ suggestions.

Our itemized responses to the each of the Reviewers’ comments follow.

Comments from the editors and reviewers:
-Reviewer 1

Comments and Suggestions for Authors

Major comments:

The manuscript needs major improvement in the English presentation

I would shorten the introduction and divide the paragraph “New insights in the TTS pathophysiology: mechanisms and pathways from current evidence” in subparagraphs: role of catecholamines, Dysregulation of metabolic pathways and Inflammation.  

1.Author comment. We thank the Reviewer 1 for his/her important suggestions, which have helped us in improving all the sections of our review. Concerning the observation on English presentation of our manuscript, significant improvements have been performed in the text of the Introduction section, subsequent paragraphs, and the conclusion section. In addition, we revised introduction, by reducing it and adding another appropriate section with short paragraphs, for describing the pathways and mechanisms involved in TTS pathophysiology, as suggested (see lines 37-124; 129-242, pgg1-6).

The figure 1 is very similar to the one of the paper referenced in 5. The authors should at least add the sentence: “modified from”

2.Author comment. As suggested, we have modified the Figure 1.

Line 94: I would suggest a possible mechanism for the increased prevalence of TTS in males: for example, the genetic background, or differences in stressful environments as compared with other populations…..

3.Author comment. We thank the Revisor 1 for the suggested mechanism, which has been reported in the sentence. Precisely, we have reported: In Japan, men are more affected than women, for reasons not completely identified [11]. However, differences in the genetic background or stressful environments of Japanese population than other populations might be the potential causes (see lines 94-97, pg 3).   

Minor comments:

Line 26: Accordingly, familial form of TTS has been described” should be: “Accordingly, familial forms of TTS have been described”.

Line 50: “it reports, that TTS also represents” should be : It has been reported that TTS may also represent”

 Line 60: “may increase the rate of vulnerability” I would add “may increase the rate of vulnerability to the infection and its complications”  

Line 63: the sentence needs the references.

Line 65: the sentence needs the references.

Line 65: “Likewise, it reports” should be: “Likewise, it has been reported”. Furthermore, the sentence needs references on line 67

Line 71: the sentence needs the references.

Line 71-72: “This has recently led to define TTS as happy heart syndrome.” Would be better as: “This has recently led to define some forms of TTS as happy heart syndromes.”

Line 74: the sentence needs the references.

Line 76: “performed thanks to” should be: “based on the evidence of…”

Lines 79-80: “apical and midventricular myocardium.” Better if “apical and midventricular Left ventricular walls”

Lines 85-87: “Thus, an interindividual heterogeneity in onset and progression characterizes TTS, as typical is the strong predominance of TTS in postmenopausal women” I would change to: “Thus, an interindividual heterogeneity in onset and progression characterizes TTS. Despite this heterogeneity, TTS is typically described in postmenopausal women”.

Lines 85-90: the sentences need the references.

Lines 93-94: “In Japan, men are, while, more affected than women, for unknown reasons” I would delete “while”

4.Author comment. We thank the Revisor 1 for all the changes above suggested. We have performed all the corrections as indicated in text in blue ( see lines 37-124; 129-242, pgg1-6).

Lines-95-98: “Concerning the TTS treatments, they have been not established by European Society of Cardiology (ESC), because of absent of a precise understanding on molecular and cellular mechanisms involved in TTS pathophysiology until now” Please

rephrase checking for grammar. Furthermore why do the authors just cite the ESC and not AHA and ACC?

5.Author comment. We have rephrased and improved grammar quality of the indicated sentences  (see lines-98-106, pg. 3).

Line 209: I would delete “Another aspect to clarify:”

6..Author comment. We have changed the title of this paragraph.

CommentsReviewers.

We thank the Reviewers for their appropriate and constructive comments. We took into great considerations the criticisms, and particularly those from reviewer 1# and we feel we addressed all the raised concerns. We believe that our manuscript is significantly improved by following the Reviewers’ suggestions.

Our itemized responses to the each of the Reviewers’ comments follow.

Comments from the editors and reviewers:
-Reviewer 1

Comments and Suggestions for Authors

Major comments:

The manuscript needs major improvement in the English presentation

I would shorten the introduction and divide the paragraph “New insights in the TTS pathophysiology: mechanisms and pathways from current evidence” in subparagraphs: role of catecholamines, Dysregulation of metabolic pathways and Inflammation.  

1.Author comment. We thank the Reviewer 1 for his/her important suggestions, which have helped us in improving all the sections of our review. Concerning the observation on English presentation of our manuscript, significant improvements have been performed in the text of the Introduction section, subsequent paragraphs, and the conclusion section. In addition, we revised introduction, by reducing it and adding another appropriate section with short paragraphs, for describing the pathways and mechanisms involved in TTS pathophysiology, as suggested (see lines 37-124; 129-242, pgg1-6).

The figure 1 is very similar to the one of the paper referenced in 5. The authors should at least add the sentence: “modified from”

2.Author comment. As suggested, we have modified the Figure 1.

Line 94: I would suggest a possible mechanism for the increased prevalence of TTS in males: for example, the genetic background, or differences in stressful environments as compared with other populations…..

3.Author comment. We thank the Revisor 1 for the suggested mechanism, which has been reported in the sentence. Precisely, we have reported: In Japan, men are more affected than women, for reasons not completely identified [11]. However, differences in the genetic background or stressful environments of Japanese population than other populations might be the potential causes (see lines 94-97, pg 3).   

Minor comments:

Line 26: Accordingly, familial form of TTS has been described” should be: “Accordingly, familial forms of TTS have been described”.

Line 50: “it reports, that TTS also represents” should be : It has been reported that TTS may also represent”

 Line 60: “may increase the rate of vulnerability” I would add “may increase the rate of vulnerability to the infection and its complications”  

Line 63: the sentence needs the references.

Line 65: the sentence needs the references.

Line 65: “Likewise, it reports” should be: “Likewise, it has been reported”. Furthermore, the sentence needs references on line 67

Line 71: the sentence needs the references.

Line 71-72: “This has recently led to define TTS as happy heart syndrome.” Would be better as: “This has recently led to define some forms of TTS as happy heart syndromes.”

Line 74: the sentence needs the references.

Line 76: “performed thanks to” should be: “based on the evidence of…”

Lines 79-80: “apical and midventricular myocardium.” Better if “apical and midventricular Left ventricular walls”

Lines 85-87: “Thus, an interindividual heterogeneity in onset and progression characterizes TTS, as typical is the strong predominance of TTS in postmenopausal women” I would change to: “Thus, an interindividual heterogeneity in onset and progression characterizes TTS. Despite this heterogeneity, TTS is typically described in postmenopausal women”.

Lines 85-90: the sentences need the references.

Lines 93-94: “In Japan, men are, while, more affected than women, for unknown reasons” I would delete “while”

4.Author comment. We thank the Revisor 1 for all the changes above suggested. We have performed all the corrections as indicated in text in blue ( see lines 37-124; 129-242, pgg1-6).

Lines-95-98: “Concerning the TTS treatments, they have been not established by European Society of Cardiology (ESC), because of absent of a precise understanding on molecular and cellular mechanisms involved in TTS pathophysiology until now” Please

rephrase checking for grammar. Furthermore why do the authors just cite the ESC and not AHA and ACC?

5.Author comment. We have rephrased and improved grammar quality of the indicated sentences  (see lines-98-106, pg. 3).

Line 209: I would delete “Another aspect to clarify:”

6..Author comment. We have changed the title of this paragraph.

CommentsReviewers.

We thank the Reviewers for their appropriate and constructive comments. We took into great considerations the criticisms, and particularly those from reviewer 1# and we feel we addressed all the raised concerns. We believe that our manuscript is significantly improved by following the Reviewers’ suggestions.

Our itemized responses to the each of the Reviewers’ comments follow.

Comments from the editors and reviewers:
-Reviewer 1

Comments and Suggestions for Authors

Major comments:

The manuscript needs major improvement in the English presentation

I would shorten the introduction and divide the paragraph “New insights in the TTS pathophysiology: mechanisms and pathways from current evidence” in subparagraphs: role of catecholamines, Dysregulation of metabolic pathways and Inflammation.  

1.Author comment. We thank the Reviewer 1 for his/her important suggestions, which have helped us in improving all the sections of our review. Concerning the observation on English presentation of our manuscript, significant improvements have been performed in the text of the Introduction section, subsequent paragraphs, and the conclusion section. In addition, we revised introduction, by reducing it and adding another appropriate section with short paragraphs, for describing the pathways and mechanisms involved in TTS pathophysiology, as suggested (see lines 37-124; 129-242, pgg1-6).

The figure 1 is very similar to the one of the paper referenced in 5. The authors should at least add the sentence: “modified from”

2.Author comment. As suggested, we have modified the Figure 1.

Line 94: I would suggest a possible mechanism for the increased prevalence of TTS in males: for example, the genetic background, or differences in stressful environments as compared with other populations…..

3.Author comment. We thank the Revisor 1 for the suggested mechanism, which has been reported in the sentence. Precisely, we have reported: In Japan, men are more affected than women, for reasons not completely identified [11]. However, differences in the genetic background or stressful environments of Japanese population than other populations might be the potential causes (see lines 94-97, pg 3).   

Minor comments:

Line 26: Accordingly, familial form of TTS has been described” should be: “Accordingly, familial forms of TTS have been described”.

Line 50: “it reports, that TTS also represents” should be : It has been reported that TTS may also represent”

 Line 60: “may increase the rate of vulnerability” I would add “may increase the rate of vulnerability to the infection and its complications”  

Line 63: the sentence needs the references.

Line 65: the sentence needs the references.

Line 65: “Likewise, it reports” should be: “Likewise, it has been reported”. Furthermore, the sentence needs references on line 67

Line 71: the sentence needs the references.

Line 71-72: “This has recently led to define TTS as happy heart syndrome.” Would be better as: “This has recently led to define some forms of TTS as happy heart syndromes.”

Line 74: the sentence needs the references.

Line 76: “performed thanks to” should be: “based on the evidence of…”

Lines 79-80: “apical and midventricular myocardium.” Better if “apical and midventricular Left ventricular walls”

Lines 85-87: “Thus, an interindividual heterogeneity in onset and progression characterizes TTS, as typical is the strong predominance of TTS in postmenopausal women” I would change to: “Thus, an interindividual heterogeneity in onset and progression characterizes TTS. Despite this heterogeneity, TTS is typically described in postmenopausal women”.

Lines 85-90: the sentences need the references.

Lines 93-94: “In Japan, men are, while, more affected than women, for unknown reasons” I would delete “while”

4.Author comment. We thank the Revisor 1 for all the changes above suggested. We have performed all the corrections as indicated in text in blue ( see lines 37-124; 129-242, pgg1-6).

Lines-95-98: “Concerning the TTS treatments, they have been not established by European Society of Cardiology (ESC), because of absent of a precise understanding on molecular and cellular mechanisms involved in TTS pathophysiology until now” Please

rephrase checking for grammar. Furthermore why do the authors just cite the ESC and not AHA and ACC?

5.Author comment. We have rephrased and improved grammar quality of the indicated sentences  (see lines-98-106, pg. 3).

Line 209: I would delete “Another aspect to clarify:”

6..Author comment. We have changed the title of this paragraph.

CommentsReviewers.

We thank the Reviewers for their appropriate and constructive comments. We took into great considerations the criticisms, and particularly those from reviewer 1# and we feel we addressed all the raised concerns. We believe that our manuscript is significantly improved by following the Reviewers’ suggestions.

Our itemized responses to the each of the Reviewers’ comments follow.

Comments from the editors and reviewers:
-Reviewer 1

Comments and Suggestions for Authors

Major comments:

The manuscript needs major improvement in the English presentation

I would shorten the introduction and divide the paragraph “New insights in the TTS pathophysiology: mechanisms and pathways from current evidence” in subparagraphs: role of catecholamines, Dysregulation of metabolic pathways and Inflammation.  

1.Author comment. We thank the Reviewer 1 for his/her important suggestions, which have helped us in improving all the sections of our review. Concerning the observation on English presentation of our manuscript, significant improvements have been performed in the text of the Introduction section, subsequent paragraphs, and the conclusion section. In addition, we revised introduction, by reducing it and adding another appropriate section with short paragraphs, for describing the pathways and mechanisms involved in TTS pathophysiology, as suggested (see lines 37-124; 129-242, pgg1-6).

The figure 1 is very similar to the one of the paper referenced in 5. The authors should at least add the sentence: “modified from”

2.Author comment. As suggested, we have modified the Figure 1.

Line 94: I would suggest a possible mechanism for the increased prevalence of TTS in males: for example, the genetic background, or differences in stressful environments as compared with other populations…..

3.Author comment. We thank the Revisor 1 for the suggested mechanism, which has been reported in the sentence. Precisely, we have reported: In Japan, men are more affected than women, for reasons not completely identified [11]. However, differences in the genetic background or stressful environments of Japanese population than other populations might be the potential causes (see lines 94-97, pg 3).   

Minor comments:

Line 26: Accordingly, familial form of TTS has been described” should be: “Accordingly, familial forms of TTS have been described”.

Line 50: “it reports, that TTS also represents” should be : It has been reported that TTS may also represent”

 Line 60: “may increase the rate of vulnerability” I would add “may increase the rate of vulnerability to the infection and its complications”  

Line 63: the sentence needs the references.

Line 65: the sentence needs the references.

Line 65: “Likewise, it reports” should be: “Likewise, it has been reported”. Furthermore, the sentence needs references on line 67

Line 71: the sentence needs the references.

Line 71-72: “This has recently led to define TTS as happy heart syndrome.” Would be better as: “This has recently led to define some forms of TTS as happy heart syndromes.”

Line 74: the sentence needs the references.

Line 76: “performed thanks to” should be: “based on the evidence of…”

Lines 79-80: “apical and midventricular myocardium.” Better if “apical and midventricular Left ventricular walls”

Lines 85-87: “Thus, an interindividual heterogeneity in onset and progression characterizes TTS, as typical is the strong predominance of TTS in postmenopausal women” I would change to: “Thus, an interindividual heterogeneity in onset and progression characterizes TTS. Despite this heterogeneity, TTS is typically described in postmenopausal women”.

Lines 85-90: the sentences need the references.

Lines 93-94: “In Japan, men are, while, more affected than women, for unknown reasons” I would delete “while”

4.Author comment. We thank the Revisor 1 for all the changes above suggested. We have performed all the corrections as indicated in text in blue ( see lines 37-124; 129-242, pgg1-6).

Lines-95-98: “Concerning the TTS treatments, they have been not established by European Society of Cardiology (ESC), because of absent of a precise understanding on molecular and cellular mechanisms involved in TTS pathophysiology until now” Please

rephrase checking for grammar. Furthermore why do the authors just cite the ESC and not AHA and ACC?

5.Author comment. We have rephrased and improved grammar quality of the indicated sentences  (see lines-98-106, pg. 3).

Line 209: I would delete “Another aspect to clarify:”

6..Author comment. We have changed the title of this paragraph.

CommentsReviewers.

We thank the Reviewers for their appropriate and constructive comments. We took into great considerations the criticisms, and particularly those from reviewer 1# and we feel we addressed all the raised concerns. We believe that our manuscript is significantly improved by following the Reviewers’ suggestions.

Our itemized responses to the each of the Reviewers’ comments follow.

Comments from the editors and reviewers:
-Reviewer 1

Comments and Suggestions for Authors

Major comments:

The manuscript needs major improvement in the English presentation

I would shorten the introduction and divide the paragraph “New insights in the TTS pathophysiology: mechanisms and pathways from current evidence” in subparagraphs: role of catecholamines, Dysregulation of metabolic pathways and Inflammation.  

1.Author comment. We thank the Reviewer 1 for his/her important suggestions, which have helped us in improving all the sections of our review. Concerning the observation on English presentation of our manuscript, significant improvements have been performed in the text of the Introduction section, subsequent paragraphs, and the conclusion section. In addition, we revised introduction, by reducing it and adding another appropriate section with short paragraphs, for describing the pathways and mechanisms involved in TTS pathophysiology, as suggested (see lines 37-124; 129-242, pgg1-6).

The figure 1 is very similar to the one of the paper referenced in 5. The authors should at least add the sentence: “modified from”

2.Author comment. As suggested, we have modified the Figure 1.

Line 94: I would suggest a possible mechanism for the increased prevalence of TTS in males: for example, the genetic background, or differences in stressful environments as compared with other populations…..

3.Author comment. We thank the Revisor 1 for the suggested mechanism, which has been reported in the sentence. Precisely, we have reported: In Japan, men are more affected than women, for reasons not completely identified [11]. However, differences in the genetic background or stressful environments of Japanese population than other populations might be the potential causes (see lines 94-97, pg 3).   

Minor comments:

Line 26: Accordingly, familial form of TTS has been described” should be: “Accordingly, familial forms of TTS have been described”.

Line 50: “it reports, that TTS also represents” should be : It has been reported that TTS may also represent”

 Line 60: “may increase the rate of vulnerability” I would add “may increase the rate of vulnerability to the infection and its complications”  

Line 63: the sentence needs the references.

Line 65: the sentence needs the references.

Line 65: “Likewise, it reports” should be: “Likewise, it has been reported”. Furthermore, the sentence needs references on line 67

Line 71: the sentence needs the references.

Line 71-72: “This has recently led to define TTS as happy heart syndrome.” Would be better as: “This has recently led to define some forms of TTS as happy heart syndromes.”

Line 74: the sentence needs the references.

Line 76: “performed thanks to” should be: “based on the evidence of…”

Lines 79-80: “apical and midventricular myocardium.” Better if “apical and midventricular Left ventricular walls”

Lines 85-87: “Thus, an interindividual heterogeneity in onset and progression characterizes TTS, as typical is the strong predominance of TTS in postmenopausal women” I would change to: “Thus, an interindividual heterogeneity in onset and progression characterizes TTS. Despite this heterogeneity, TTS is typically described in postmenopausal women”.

Lines 85-90: the sentences need the references.

Lines 93-94: “In Japan, men are, while, more affected than women, for unknown reasons” I would delete “while”

4.Author comment. We thank the Revisor 1 for all the changes above suggested. We have performed all the corrections as indicated in text in blue ( see lines 37-124; 129-242, pgg1-6).

Lines-95-98: “Concerning the TTS treatments, they have been not established by European Society of Cardiology (ESC), because of absent of a precise understanding on molecular and cellular mechanisms involved in TTS pathophysiology until now” Please

rephrase checking for grammar. Furthermore why do the authors just cite the ESC and not AHA and ACC?

5.Author comment. We have rephrased and improved grammar quality of the indicated sentences  (see lines-98-106, pg. 3).

Line 209: I would delete “Another aspect to clarify:”

6..Author comment. We have changed the title of this paragraph.

CommentsReviewers.

We thank the Reviewers for their appropriate and constructive comments. We took into great considerations the criticisms, and particularly those from reviewer 1# and we feel we addressed all the raised concerns. We believe that our manuscript is significantly improved by following the Reviewers’ suggestions.

Our itemized responses to the each of the Reviewers’ comments follow.

Comments from the editors and reviewers:
-Reviewer 1

Comments and Suggestions for Authors

Major comments:

The manuscript needs major improvement in the English presentation

I would shorten the introduction and divide the paragraph “New insights in the TTS pathophysiology: mechanisms and pathways from current evidence” in subparagraphs: role of catecholamines, Dysregulation of metabolic pathways and Inflammation.  

1.Author comment. We thank the Reviewer 1 for his/her important suggestions, which have helped us in improving all the sections of our review. Concerning the observation on English presentation of our manuscript, significant improvements have been performed in the text of the Introduction section, subsequent paragraphs, and the conclusion section. In addition, we revised introduction, by reducing it and adding another appropriate section with short paragraphs, for describing the pathways and mechanisms involved in TTS pathophysiology, as suggested (see lines 37-124; 129-242, pgg1-6).

The figure 1 is very similar to the one of the paper referenced in 5. The authors should at least add the sentence: “modified from”

2.Author comment. As suggested, we have modified the Figure 1.

Line 94: I would suggest a possible mechanism for the increased prevalence of TTS in males: for example, the genetic background, or differences in stressful environments as compared with other populations…..

3.Author comment. We thank the Revisor 1 for the suggested mechanism, which has been reported in the sentence. Precisely, we have reported: In Japan, men are more affected than women, for reasons not completely identified [11]. However, differences in the genetic background or stressful environments of Japanese population than other populations might be the potential causes (see lines 94-97, pg 3).   

Minor comments:

Line 26: Accordingly, familial form of TTS has been described” should be: “Accordingly, familial forms of TTS have been described”.

Line 50: “it reports, that TTS also represents” should be : It has been reported that TTS may also represent”

 Line 60: “may increase the rate of vulnerability” I would add “may increase the rate of vulnerability to the infection and its complications”  

Line 63: the sentence needs the references.

Line 65: the sentence needs the references.

Line 65: “Likewise, it reports” should be: “Likewise, it has been reported”. Furthermore, the sentence needs references on line 67

Line 71: the sentence needs the references.

Line 71-72: “This has recently led to define TTS as happy heart syndrome.” Would be better as: “This has recently led to define some forms of TTS as happy heart syndromes.”

Line 74: the sentence needs the references.

Line 76: “performed thanks to” should be: “based on the evidence of…”

Lines 79-80: “apical and midventricular myocardium.” Better if “apical and midventricular Left ventricular walls”

Lines 85-87: “Thus, an interindividual heterogeneity in onset and progression characterizes TTS, as typical is the strong predominance of TTS in postmenopausal women” I would change to: “Thus, an interindividual heterogeneity in onset and progression characterizes TTS. Despite this heterogeneity, TTS is typically described in postmenopausal women”.

Lines 85-90: the sentences need the references.

Lines 93-94: “In Japan, men are, while, more affected than women, for unknown reasons” I would delete “while”

4.Author comment. We thank the Revisor 1 for all the changes above suggested. We have performed all the corrections as indicated in text in blue ( see lines 37-124; 129-242, pgg1-6).

Lines-95-98: “Concerning the TTS treatments, they have been not established by European Society of Cardiology (ESC), because of absent of a precise understanding on molecular and cellular mechanisms involved in TTS pathophysiology until now” Please

rephrase checking for grammar. Furthermore why do the authors just cite the ESC and not AHA and ACC?

5.Author comment. We have rephrased and improved grammar quality of the indicated sentences  (see lines-98-106, pg. 3).

Line 209: I would delete “Another aspect to clarify:”

6..Author comment. We have changed the title of this paragraph.

CommentsReviewers.

We thank the Reviewers for their appropriate and constructive comments. We took into great considerations the criticisms, and particularly those from reviewer 1# and we feel we addressed all the raised concerns. We believe that our manuscript is significantly improved by following the Reviewers’ suggestions.

Our itemized responses to the each of the Reviewers’ comments follow.

Comments from the editors and reviewers:
-Reviewer 1

Comments and Suggestions for Authors

Major comments:

The manuscript needs major improvement in the English presentation

I would shorten the introduction and divide the paragraph “New insights in the TTS pathophysiology: mechanisms and pathways from current evidence” in subparagraphs: role of catecholamines, Dysregulation of metabolic pathways and Inflammation.  

1.Author comment. We thank the Reviewer 1 for his/her important suggestions, which have helped us in improving all the sections of our review. Concerning the observation on English presentation of our manuscript, significant improvements have been performed in the text of the Introduction section, subsequent paragraphs, and the conclusion section. In addition, we revised introduction, by reducing it and adding another appropriate section with short paragraphs, for describing the pathways and mechanisms involved in TTS pathophysiology, as suggested (see lines 37-124; 129-242, pgg1-6).

The figure 1 is very similar to the one of the paper referenced in 5. The authors should at least add the sentence: “modified from”

2.Author comment. As suggested, we have modified the Figure 1.

Line 94: I would suggest a possible mechanism for the increased prevalence of TTS in males: for example, the genetic background, or differences in stressful environments as compared with other populations…..

3.Author comment. We thank the Revisor 1 for the suggested mechanism, which has been reported in the sentence. Precisely, we have reported: In Japan, men are more affected than women, for reasons not completely identified [11]. However, differences in the genetic background or stressful environments of Japanese population than other populations might be the potential causes (see lines 94-97, pg 3).   

Minor comments:

Line 26: Accordingly, familial form of TTS has been described” should be: “Accordingly, familial forms of TTS have been described”.

Line 50: “it reports, that TTS also represents” should be : It has been reported that TTS may also represent”

 Line 60: “may increase the rate of vulnerability” I would add “may increase the rate of vulnerability to the infection and its complications”  

Line 63: the sentence needs the references.

Line 65: the sentence needs the references.

Line 65: “Likewise, it reports” should be: “Likewise, it has been reported”. Furthermore, the sentence needs references on line 67

Line 71: the sentence needs the references.

Line 71-72: “This has recently led to define TTS as happy heart syndrome.” Would be better as: “This has recently led to define some forms of TTS as happy heart syndromes.”

Line 74: the sentence needs the references.

Line 76: “performed thanks to” should be: “based on the evidence of…”

Lines 79-80: “apical and midventricular myocardium.” Better if “apical and midventricular Left ventricular walls”

Lines 85-87: “Thus, an interindividual heterogeneity in onset and progression characterizes TTS, as typical is the strong predominance of TTS in postmenopausal women” I would change to: “Thus, an interindividual heterogeneity in onset and progression characterizes TTS. Despite this heterogeneity, TTS is typically described in postmenopausal women”.

Lines 85-90: the sentences need the references.

Lines 93-94: “In Japan, men are, while, more affected than women, for unknown reasons” I would delete “while”

4.Author comment. We thank the Revisor 1 for all the changes above suggested. We have performed all the corrections as indicated in text in blue ( see lines 37-124; 129-242, pgg1-6).

Lines-95-98: “Concerning the TTS treatments, they have been not established by European Society of Cardiology (ESC), because of absent of a precise understanding on molecular and cellular mechanisms involved in TTS pathophysiology until now” Please

rephrase checking for grammar. Furthermore why do the authors just cite the ESC and not AHA and ACC?

5.Author comment. We have rephrased and improved grammar quality of the indicated sentences  (see lines-98-106, pg. 3).

Line 209: I would delete “Another aspect to clarify:”

6..Author comment. We have changed the title of this paragraph.

CommentsReviewers.

We thank the Reviewers for their appropriate and constructive comments. We took into great considerations the criticisms, and particularly those from reviewer 1# and we feel we addressed all the raised concerns. We believe that our manuscript is significantly improved by following the Reviewers’ suggestions.

Our itemized responses to the each of the Reviewers’ comments follow.

Comments from the editors and reviewers:
-Reviewer 1

Comments and Suggestions for Authors

Major comments:

The manuscript needs major improvement in the English presentation

I would shorten the introduction and divide the paragraph “New insights in the TTS pathophysiology: mechanisms and pathways from current evidence” in subparagraphs: role of catecholamines, Dysregulation of metabolic pathways and Inflammation.  

1.Author comment. We thank the Reviewer 1 for his/her important suggestions, which have helped us in improving all the sections of our review. Concerning the observation on English presentation of our manuscript, significant improvements have been performed in the text of the Introduction section, subsequent paragraphs, and the conclusion section. In addition, we revised introduction, by reducing it and adding another appropriate section with short paragraphs, for describing the pathways and mechanisms involved in TTS pathophysiology, as suggested (see lines 37-124; 129-242, pgg1-6).

The figure 1 is very similar to the one of the paper referenced in 5. The authors should at least add the sentence: “modified from”

2.Author comment. As suggested, we have modified the Figure 1.

Line 94: I would suggest a possible mechanism for the increased prevalence of TTS in males: for example, the genetic background, or differences in stressful environments as compared with other populations…..

3.Author comment. We thank the Revisor 1 for the suggested mechanism, which has been reported in the sentence. Precisely, we have reported: In Japan, men are more affected than women, for reasons not completely identified [11]. However, differences in the genetic background or stressful environments of Japanese population than other populations might be the potential causes (see lines 94-97, pg 3).   

Minor comments:

Line 26: Accordingly, familial form of TTS has been described” should be: “Accordingly, familial forms of TTS have been described”.

Line 50: “it reports, that TTS also represents” should be : It has been reported that TTS may also represent”

 Line 60: “may increase the rate of vulnerability” I would add “may increase the rate of vulnerability to the infection and its complications”  

Line 63: the sentence needs the references.

Line 65: the sentence needs the references.

Line 65: “Likewise, it reports” should be: “Likewise, it has been reported”. Furthermore, the sentence needs references on line 67

Line 71: the sentence needs the references.

Line 71-72: “This has recently led to define TTS as happy heart syndrome.” Would be better as: “This has recently led to define some forms of TTS as happy heart syndromes.”

Line 74: the sentence needs the references.

Line 76: “performed thanks to” should be: “based on the evidence of…”

Lines 79-80: “apical and midventricular myocardium.” Better if “apical and midventricular Left ventricular walls”

Lines 85-87: “Thus, an interindividual heterogeneity in onset and progression characterizes TTS, as typical is the strong predominance of TTS in postmenopausal women” I would change to: “Thus, an interindividual heterogeneity in onset and progression characterizes TTS. Despite this heterogeneity, TTS is typically described in postmenopausal women”.

Lines 85-90: the sentences need the references.

Lines 93-94: “In Japan, men are, while, more affected than women, for unknown reasons” I would delete “while”

4.Author comment. We thank the Revisor 1 for all the changes above suggested. We have performed all the corrections as indicated in text in blue ( see lines 37-124; 129-242, pgg1-6).

Lines-95-98: “Concerning the TTS treatments, they have been not established by European Society of Cardiology (ESC), because of absent of a precise understanding on molecular and cellular mechanisms involved in TTS pathophysiology until now” Please

rephrase checking for grammar. Furthermore why do the authors just cite the ESC and not AHA and ACC?

5.Author comment. We have rephrased and improved grammar quality of the indicated sentences  (see lines-98-106, pg. 3).

Line 209: I would delete “Another aspect to clarify:”

6..Author comment. We have changed the title of this paragraph.

Reviewer 2 Report

I still suggest that the work is not interesting for the readers of the journal

Author Response

Comments and Suggestions for Authors

 I still suggest that the work is not interesting for the readers of the journal

1.Author comment. We still thank the Reviewer 2  for his/her evaluation, even if negative, and the time dedicated in reading our manuscript. However, ulterior changes, improvements have been performed. I hope in the change of his/her opinion.

Round 3

Reviewer 1 Report

The manuscript looks much better now; however I would suggest to check one more time for further minor improvements of the English presentation.